# Use of GIS and Remote Sensing Data to Understand the Impacts of Land Use/Land Cover Changes (LULCC) on Snow Leopard (*Panthera uncia*) Habitat in Pakistan

Tauheed Ullah Khan [1,2], Abdul Mannan [3], Charlotte E. Hacker [4], Shahid Ahmad [5], Muhammad Amir Siddique [6], Barkat Ullah Khan [7], Emad Ud Din [8], Minhao Chen [1], Chao Zhang [1], Moazzam Nizami [9] and Xiaofeng Luan [1,*]

1   School of Ecology and Nature Conservation, Beijing Forestry University, Beijing 100083, China; eco.tauheed@hotmail.com (T.U.K.); demominhao@gmail.com (M.C.); zc1192@163.com (C.Z.)
2   Department of Zoology, Kohat University, Kohat 26170, Pakistan
3   Department of Forestry, Karakoram International University, Gilgit-Baltistan 15100, Pakistan; abdul.mannan@kiu.edu.pk
4   Department of Biological Sciences, Duquesne University, Pittsburgh, PA 15282, USA; hackerc@duq.edu
5   Key Laboratory of Animal Physiology, Biochemistry and Molecular Biology of Hebei Province, College of Life Sciences, Hebei Normal University, Shijiazhuang 050016, China; shahidbuneri87@hotmail.com
6   School of Landscape Architecture, Tianjin University, Tianjin 300384, China; amir@tju.edu.cn
7   Ministry of Climate Change Islamabad, Islamabad 44000, Pakistan; barkat.paki@gmail.com
8   College of Environmental Science and Engineering, Beijing Forestry University, Beijing 100083, China; thegreatemi@yahoo.com
9   Department of Forestry and Wildlife Management, University of Haripur, Haripur 26100, Pakistan; moazzam.nizami@uoh.edu.pk
*   Correspondence: luanxiaofeng@bjfu.edu.cn; Tel./Fax: +86-139-1009-0393

**Abstract:** Habitat degradation and species range contraction due to land use/land cover changes (LULCC) is a major threat to global biodiversity. The ever-growing human population has trespassed deep into the natural habitat of many species via the expansion of agricultural lands and infrastructural development. Carnivore species are particularly at risk, as they demand conserved and well-connected habitat with minimum to no anthropogenic disturbance. In Pakistan, the snow leopard (*Panthera uncia*) is found in three mountain ranges—the Himalayas, Hindukush, and Karakoram. Despite this being one of the harshest environments on the planet, a large population of humans reside here and exploit surrounding natural resources to meet their needs. Keeping in view this exponentially growing population and its potential impacts on at-risk species like the snow leopard, we used geographic information systems (GIS) and remote sensing with the aim of identifying and quantifying LULCC across snow leopard range in Pakistan for the years 2000, 2010, and 2020. A massive expansion of 1804.13 km$^2$ (163%) was observed in the built-up area during the study period. Similarly, an increase of 3177.74 km$^2$ (153%) was observed in agricultural land. Barren mountain land increased by 12,368.39 km$^2$ (28%) while forest land decreased by 2478.43 km$^2$ (28%) and area with snow cover decreased by 14,799.83 km$^2$ (52%). Drivers of these large-scale changes are likely the expanding human population and climate change. The overall quality and quantity of snow leopard habitat in Pakistan has drastically changed in the last 20 years and could be compromised. Swift and direct conservation actions to monitor LULCC are recommended to reduce any associated negative impacts on species preservation efforts. In the future, a series of extensive field surveys and studies should be carried out to monitor key drivers of LULCC across the observed area.

**Keywords:** snow leopard range; anthropogenic disturbance; infrastructure development; habitat quality; carnivore; northern Pakistan

## 1. Introduction

The rapidly decreasing numbers of wildlife species at local, regional, and global scales can largely be attributed to land-use and land-cover changes (LULCC). LULCC are structural modifications made by humans on Earth's terrestrial surface [1]. They can cause habitat destruction, alteration, and fragmentation. Land use is the utilization of land surface by humans to meet their needs, such as food procurement, pastoralism, agricultural development, construction of residential homes and economic zones, as well as industrial infrastructure. Land cover is the natural appearance, outlook, and biophysical characteristics of a land surface, such as forests, deserts, and grasslands. Several studies have concluded that LULCC activities are intensifying, and that wildlife habitat is increasingly being developed for agriculture and infrastructure [2,3]. Settlement development and the associated expansion of transportation networks are among the most influential LULCC affecting current species spatial distributions and habitat continuity [4,5]. Studies have also shown further expansion of already built-up areas into natural habitats [6,7]. This encroachment may ultimately impact conservation hotspots, even if they are located far from urban centers [8].

The conversion of wildlife habitat to land for agriculture or infrastructure has contributed to global biodiversity loss [3]. Humans have modified approximately 50% of natural land surfaces to artificial forms and shapes [9]. The ever-growing needs of the rising human population has intensified the rate of LULCC, driving unprecedented shifts in ecosystems at local, regional, and global scales [10,11]. LULCC can impact a variety of factors including climate, precipitation, vegetation cover, land surface temperature and community structure and composition [12–15]. Terrestrial landscapes are increasingly subject to human alteration and associated LULCC changes, which has proven lethal to global biodiversity [3,16–20].

LULCC have been directly linked to altering wildlife species distributions, causing habitat modification and, ultimately, population decline or extinction [21–23]. It can also affect the daily movement and seasonal migration of many species [24]. [25] reported that populations of forest dwelling species declined exponentially when their habitat was fragmented or reduced due to LULCC. The study suggested that alteration of native habitat into agricultural or infrastructural land not only caused a decline in species population, but also increased the intensity of conflicts between humans and wildlife [25]. According to the IUCN Red List (IUCN 2017), 46% of species face extinction risks attributed to land-use change. The nature and intensity of LULCC effects vary from species to species, with some being more at-risk to its negative impacts than others [13,26–28]. For example, most species of the order Carnivora have slow population growth rates and low population sizes [29]. They also require large areas to acquire food and shelter, and to find mates [30]. These factors make the negative impacts of LULCC much more severe. In the previous two centuries, the population and diversity of terrestrial mammalian carnivores has declined by 95% to 99% in many parts of the world [31]. This has largely been due to human related factors, of which LULCC ranks towards the top.

Snow leopards are one such carnivore species facing reduced distribution ranges because of human activities through direct habitat alteration and exploitation in the form of LULCC [32]. Habitat thinning and loss is one of the major threats to this iconic, rare, and stunning species [33]. The snow leopard meta-population is found sparsely distributed across the rugged mountains [34] of 12 countries in Central Asia, including Afghanistan, Bhutan, China, India, Kazakhstan, Kyrgyzstan, Mongolia, Nepal, Pakistan, Russia, Tajikistan, and Uzbekistan [35–38]. A small area of potential snow leopard habitat was reported in Myanmar [39] though presence of the species has yet to be confirmed [37].

This remarkable and elusive species serves key roles in ecosystem function as a top predator and can serve as an indicator of overall high-altitude ecosystem health [40]. As an umbrella species, if the snow leopard population thrives, so will countless other sympatric species [41]. This includes humans, as millions of people depend on river systems tied to mountain ecosystems. Unfortunately, the low, sparsely distributed, and declining

population of snow leopards suggest that the species is not thriving. The estimated global snow leopard population is approximately 2710 to 3386 mature individuals and decreasing [37]. Among other threats, habitat loss, degradation, and fragmentation due to LULCC are depleting their future [42]. Humans are exploiting available wildlife habitat and developing it for agriculture and infrastructure. Pastoralism and livestock numbers are also growing, with grazing grounds pushing further into snow leopard habitat as grassland degrades [35,43]. In recent decades, new roads and mines have encroached on their remaining range [44]. Despite the clear threat of LULCC to the snow leopard, very little is known about the status of LULCC within species range.

To better understand the pattern, magnitude, and consequences of LULCC on a species, it is necessary to have accurate information regarding the previous (Past) and recent (Present) LULCC and land classes. LULCC can be assessed using various methods. Geographic information systems (GIS) and remote sensing have been introduced in the field of conservation to calculate the magnitude of LULCC of any surface of any size and shape. Remote sensing is the science of obtaining information about objects or areas from a distance, typically from an aircraft or a satellite. It involves a process in which the physical characteristics of an area are monitored and detected by measuring the radiation reflected and emitted by that area. The resulting multi spectral satellite images (MSI) have been used effectively by many researchers for detecting and measuring the spatial and temporal dynamics of forest cover change [45]. GIS and remote sensing data across periodic intervals provide information for land use change analysis, modeling, and management. It is a cost-effective, rapid and an accurate method of LULCC measurement [46].

Most of the snow leopard's range in Pakistan lies in a climate change-sensitive, ecologically rich, and fragile area. Pakistan supports the third largest snow leopard population (250–400 individuals, tied with India) throughout its 12-country range with a total estimated area of about 80,000 km$^2$ [47], half of which is considered prime habitat [48]. However, a recently published study concluded that a large proportion of snow leopard range in Pakistan consists of very low-quality habitat [49]. Pakistan is the sixth most populous country in the world [50,51] and is fifth on the list of countries most vulnerable to climate change according to the 2020 Global Climate Risk Index [52]. The aim of this study was to identify and quantify LULCC across snow leopard range in Pakistan in an effort to better understand how these changes may impact snow leopard populations and their surrounding ecosystems. We hypothesized that increasing human populations within snow leopard habitat would be linked to changes in LULCC with climate change playing an additional role in LULCC.

## 2. Materials and Methods

### 2.1. Study Area

The snow leopard in Pakistan can be found in the Himalaya, Karakoram, Pamir, and Hindu Kush mountain ranges. These span across three administrative units, including Gilgit-Baltistan (GB), Khyber Pakhtunkhwa (KP), and Azad Jammu Kashmir (AJK). These magnificent mountain ranges have many precipitous peaks and glaciers. Despite having harsh climatic conditions and inhospitable geographic features, these areas are heavily inhabited by people (population estimate: 6,815,772) (Figure 1). Snow leopard habitat characteristics include steep, rugged, and broken terrain and rocky outcrops, forest, shrubland, and grassland (e.g., inland cliffs, and mountain peaks) [53]. Mountain ridges, cliff edges, and well-defined drainage lines serve as common travel routes and sites for the deposition of signs, including scrapes, scats, and scent marks [54]. Protected areas have an area totaling approximately 2118 km$^2$, which is about 24% of total snow leopard range in our study area (Figure 1).

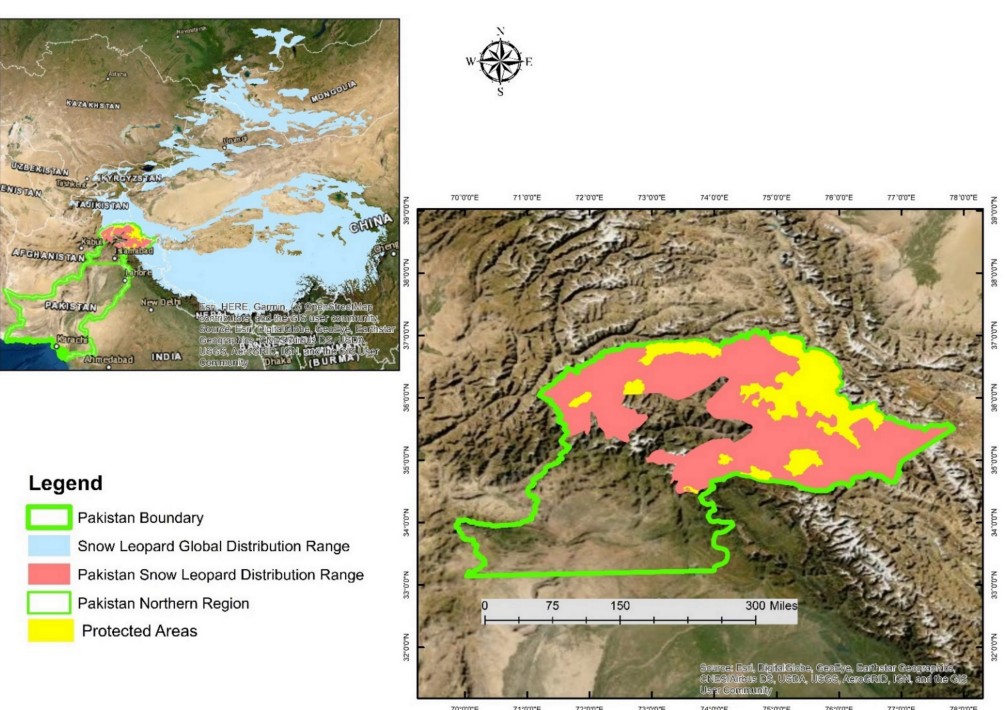

**Figure 1.** Location of the Study Area.

The area has a very rich and unique diversity of flora and fauna. About 90% of the country's natural forests are spread in this region. It has a moist temperate zone in the western Himalayas and a semi-arid environment in the northern Karakorum and Hindu Kush. Four vegetation zones including permanent snowfields, subalpine scrub zones, alpine dry steppes and alpine meadows can be identified in the area. In addition to the snow leopard, many other rare and ecologically important carnivores like the brown bear (*Ursus arctos*), Asiatic black bear (*Ursus thibetanus*), grey wolf (*Canis lupus*), Eurasian lynx (*Lynx lynx*), Pallas's cat (*Otocolobus manual*), and wild ungulates including the flare-horned markhor (*Capra falconeri cashmirensis*), Ladakh urial (*Ovis orientalis vignei*), musk deer (*Moschus chrysogaster*), blue sheep (*Pseudois nayaur*), Himalayan Ibex (*Capra ibex sibirica*), Marco Polo sheep (*Ovis ammon polii*), and woolly flying squirrel (*Eupetaurus cinereus*) are also found here.

*2.2. Data Collection*

Snow leopard distribution range as determined for the 2017 IUCN assessment was downloaded from the IUCN's Red List of Threatened Species website (http://www.iucnredlist.org/technical-documents/spatial-data) as a shapefile document (IUCN Red List, accessed on 5 January 2021) (Figure 1). Range borders were based on a gathering of experts in 2008 with data that supported a total range size of 2.8 million km$^2$ [55]. The snow leopard's range for Pakistan was extracted by using the clipping option in ArcGIS version 10.2 (Environmental Systems Research Institute, Redlands, CA, USA). For monitoring LULCC, it is necessary to have data from at least two time periods for comparison. For this study, the Landsat 5 Thematic Mapper (TM) and Landsat 8 Operational Land Imager (OLI) satellite images were downloaded from the United States Geological Survey (USGS) website (https://glovis.usgs.gov) (accessed on 7 January 2021). In this study, we wanted to observe LULCC over the last twenty years. To measure an observable change, we carried out analysis with two gaps of ten years and images were downloaded for the years 2000, 2010, and 2020. The Landsat 5 TM had seven spectral bands with a spatial resolution of 30 m, Landsat 8 (OLI) had 9 bands, where band 1 to 7 and band 9 had a spatial resolution of 30 m. While a panchromatic band with a spatial resolution of 15 m. Images were downloaded for the months of April and October for each respective year to obtain

cloud free or nearly cloud free images for the mentioned study periods [54,55]. Images with minimum or no cloud cover (0–10%) were selected and downloaded for the study period. The details of the data sources and the sensor specifications are given in (Table 1). The data used for the LULCC analysis included the satellite images of the entire study area for the study years, the Digital Elevation Model (DEM), as well as topographic sheets, road maps, and settlement maps of the study area.

**Table 1.** Information surrounding data sources and satellite sensor specifications.

| Sensor | Year | Resolution | Bands | Source |
|---|---|---|---|---|
| Landsat 5 TM | 2000 | 30 m | Multispectral | USGS Glovis |
| Landsat 5 TM | 2010 | 30 m | Multispectral | USGS Glovis |
| Landsat 8 OLI | 2020 | 30 m | Multispectral | USGS Glovis |

2.2.1. Image Pre-Processing

The radiometric correction of the satellite images for 2000, 2010, and 2020 was done using ENVI (5.3). ENVI is a software used for processing and analyzing geospatial imagery. It is also used to identify and build classification categories and to determine accuracy of land use classes [56]. The Digital numbers (DN's) of the images were converted to radiance (Equation (1)) and reflectance using the top of atmosphere (TOA) process (Equation (2)).

$$L_\lambda = Gain \times pixel\ value + offset \tag{1}$$

$$p_\lambda = \frac{\pi \times L_\lambda \times d^2}{ESUN_\lambda \times sin\theta} \tag{2}$$

Whereas in Equations (1) and (2), $L_\lambda$ is radiance, $p_\lambda$ is TOA reflectance, $d$ is the earth-Sun distance, $ESUN_\lambda$ is solar irradiance and $\theta$ represents the elevation of the sun in degrees.

Atmospheric conditions typically differ between dates. These varying atmospheric conditions can affect spectral signatures. Therefore, the atmospheric calibration method was used to convert TOA reflectance following [56]. Fast Line-of-Sight Atmospheric Analysis of Hypercubes (FLAASH) was used to convert TOA reflectance to surface reflectance of different factors including water vapors, humidity, haze, and aerosols to eliminate or minimize their effects on image quality and performance.

2.2.2. Image Classification

Information surrounding LULCC can be obtained from multiband raster imagery through the process of image interpretation and classification [57]. Image classification (supervised or unsupervised) is intended for an automatic categorization of pixels with a common reflectance range into a specific LULCC class [58–60] Supervised classification is a user guided approach that involves the selection of training sites as a reference for categorization [61–63]. Many methods are used to generate supervised classification, such as K- nearest neighbor, minimum distance classification and Maximum likelihood classifier [64]. For the present study, we adopted the more commonly used Maximum likelihood classifier for LULCC classification using ENVI (5.3). The Maximum Likelihood algorithm quantitively evaluates variance and assigns each pixel to the classifier which has the highest possibility of association [65]. Based on field observations, scientific literature and visual assessment of high-resolution Google Earth images, the area was classified into six different classes, including built-up, forest land, barren mountains, snow cover, agricultural land, and water (Table 2) by using the supervised classification method with the maximum likelihood algorithm in ENVI (5.3). We used 40 training area polygons in

each class while the decision rule of maximum likelihood was used for image classification, as shown in Equation (3).

$$gi\,(x) = Inp(wi) - \frac{1}{2}In|\Sigma i|A = \pi r^2 - \frac{1}{2}(x-mi)^T \Sigma^{i-1}\,(x-mi) \tag{3}$$

where $g_i$ is land use class, $x$ is $n$-dimension data, n is the number of bands, $p\,(w_i)$ is the probability of class wi appeared in wi class, $|\Sigma i|$ is co-varience matrix data from $wi$ class, $\Sigma^{i-1}$ is the inverse matrix and $m_i$ is the vector.

**Table 2.** The Land Use/Land Cover Changes (LULCC) classification scheme used for the study area.

| Scheme No. | Class Name | Description |
| --- | --- | --- |
| 1 | Forest Land | Land cover with mature forest or reserved for the growth of forest |
| 2 | Built-up Area | Any artificial infrastructure, residential buildings, commercial areas, industrial zones, roads, factories, villages, towns or cities |
| 3 | Agricultural Land | Crop fields, orchids, gardens |
| 4 | Water Bodies | Rivers, streams, lakes, ponds, and other water reservoirs |
| 5 | Barren Mountains | Mountains parts having no vegetation or snow cover |
| 6 | Snow Cover | Area coverd with permanent or seasonal snow |

### 2.2.3. Accuracy Assessment

Accuracy assessment is essential for individual classifications if the data are to be valuable in change detection [66]. The comparison of classification results and reference information was completed by using a confusion matrix. In addition, a non-parametric Kappa coefficient test was performed to measure the degree of classification accuracy [67]. For the accuracy assessment of land cover maps separated from satellite images, stratified random sampling was used to represent various land cover classes of the region. A total of 300 points were extracted and imported into the classified map. The confusion matrix was run in ENVI 5.3 for accuracy assessment and to calculate the user and producer accuracies for each land-use class. The kappa statistic was calculated by observed and expected formula as described by [68] (Equation (4)). This metric provides the overall accuracy of the confusion matrix. It is calculated by dividing the total number of correct pixels (diagonal values) by the total number of pixels in the confusion matrix. According to [69] the minimum accuracy value for reliable land cover classification is 80%. However, acceptable accuracy levels may vary by application and task [68].

$$K = \frac{N\sum_{i=1}^{n}m_{i,i}\,-\,\sum_{i=1}^{n}(G_iC_i)}{N^2 - \sum_{i=1}^{n}(G_iC_i)} \tag{4}$$

### 2.2.4. Assessment of LULCC Category Interconversion

The interconversion analysis of LULCC categories were carried out in ENVI software (ver. 5.3). The graphical illustrates of the results data were developed in RStudio (4.2) using the ggplot2 and ggcorplot libraries.

## 3. Results

### 3.1. Overall LULCC Dynamics across Snow Leopard Range in Pakistan

We used remote sensing to identify the land use categories and LULCC across snow leopard range in Pakistan. The overall dynamics of LULCC in the study area for the study period are given in Table 3 and Figure 2. According to the snow leopard range shapefile downloaded from the IUCN website, a large proportion (69.50%) of snow leopard range spans across GB (59,373.47 km$^2$), followed by KP (20,866.22 km$^2$) and AJK (5195.73 km$^2$).

A massive expansion of 163% was observed for built-up area during the study period. Similarly, an 153% increase of 3177.74 km² was observed for agricultural land (Figure 3). Barren mountain increased by 28%. This increase could be due to a massive deforestation and reduction of snow cover area, as evident of continuous shrinkage in forest area during the study period. Forest land was reduced by 27% from 2000 to 2020. Snow cover area was reduced by 52%. Meanwhile, a 52% decrease was observed for water bodies during the study period.

**Table 3.** The overall dynamics of LULCC in the study area during the study period.

| Years | Forest Land (km²) | Built-Up Area (km²) | Agricultural Land (km²) | Barren Mountains (km²) | Water Bodies (km²) | Snow Cover (km²) | Total |
|---|---|---|---|---|---|---|---|
| 2000 | 9128.27 | 1103.58 | 2071.36 | 44,738.89 | 138.03 | 28,255.29 | 85,435.42 |
| 2010 | 10,292.91 | 1787.87 | 4437.78 | 45,769.80 | 522.88 | 22,624.17 | 85,435.42 |
| 2020 | 6649.84 | 2907.71 | 5249.10 | 57,107.28 | 66.02 | 13,455.46 | 85,435.42 |
| Change (km²) | −2478.43 | 1804.13 | 3177.74 | 12,368.39 | −72.00 | −14,799.83 | |
| % Change | −27 | 163 | 153 | 28 | −52 | −52 | |

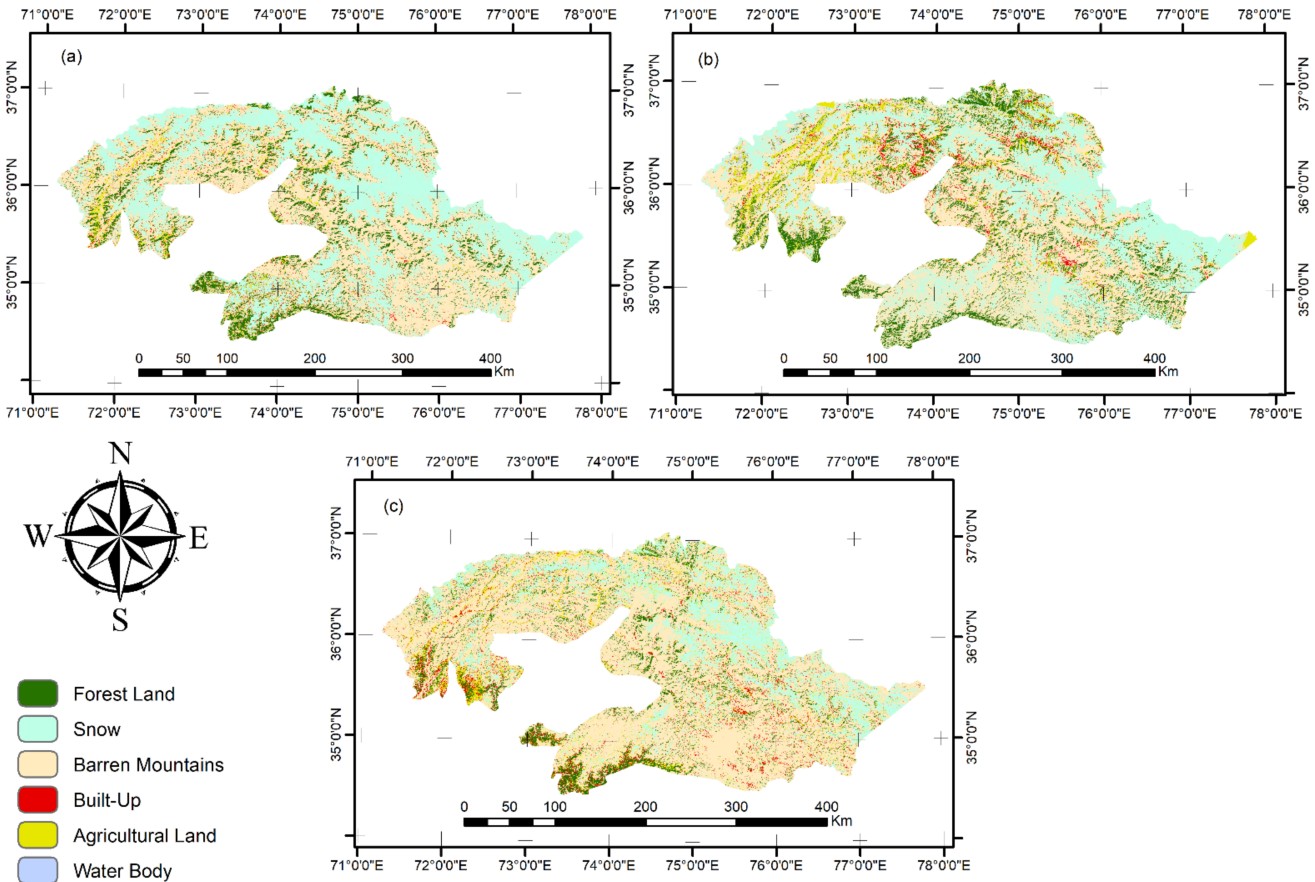

**Figure 2.** LULCC across snow leopard range in Pakistan in (**a**) 2000, (**b**) 2010, and (**c**) 2020.

### 3.2. LULCC Dynamics in Snow Leopard Range in GB

The overall dynamics of LULCC in the GB portion of Pakistan's snow leopard range is given in Table 4 and Figure 4. A loss of 32% of forest land was calculated in GB from 2000 to 2020. A massive loss of 45% was observed in snow cover area. There was an exponential increase of 151%, 316% and 25% for built-up area, agricultural land, and area of barren mountains, respectively. Water bodies in GB increased by 121%.

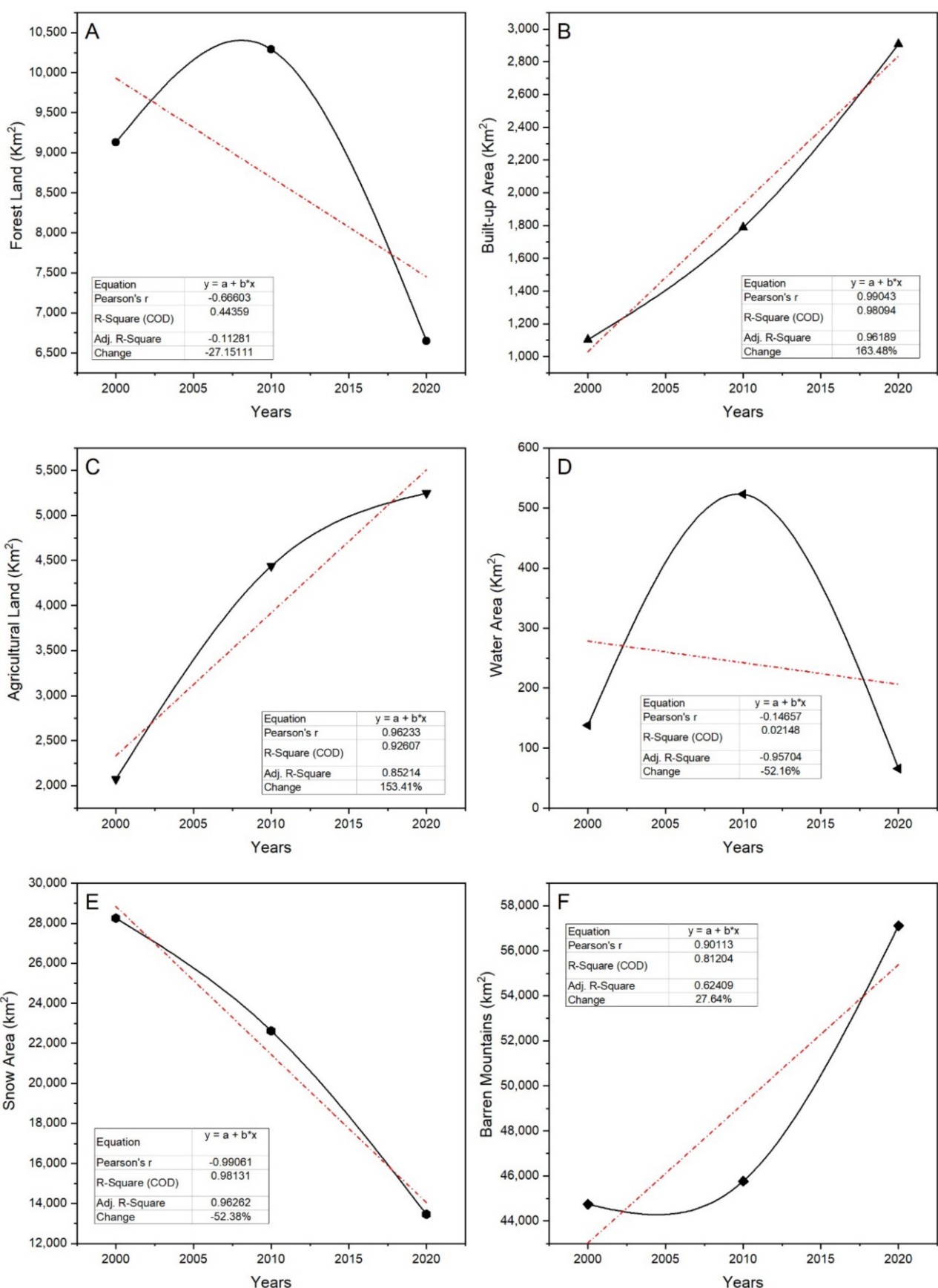

**Figure 3.** Trend of different LULCC categories in the study area during the study period. (**A**) forest land, (**B**) built-up area, (**C**) agriculture land, (**D**) water bodies, (**E**) snow-covered area, (**F)** barren mountains.

**Table 4.** LULCC Dynamics in Snow Leopard Range in Gilgit-Baltistan (GB).

| Years | Forest Land (km²) | Built up (km²) | Agricultural Land (km²) | Barren Mountains (km²) | Water Bodies (km²) | Snow Cover (km²) | Total (km²) |
|---|---|---|---|---|---|---|---|
| 2000 | 5470.32 | 590.48 | 678.57 | 32,040.24 | 24.66 | 20,569.19 | 59,373.47 |
| 2010 | 6845.54 | 1477.52 | 2067.88 | 31,499.38 | 136.56 | 17,346.59 | 59,373.47 |
| 2020 | 3693.12 | 1484.14 | 2825.16 | 40,019.07 | 54.45 | 11,296.99 | 59,372.92 |
| Change (km²) | −1777.20 | 893.66 | 2146.58 | 7978.82 | 29.79 | −9272.20 | |
| % Change | −32 | 151 | 316 | 25 | 121 | −45 | |

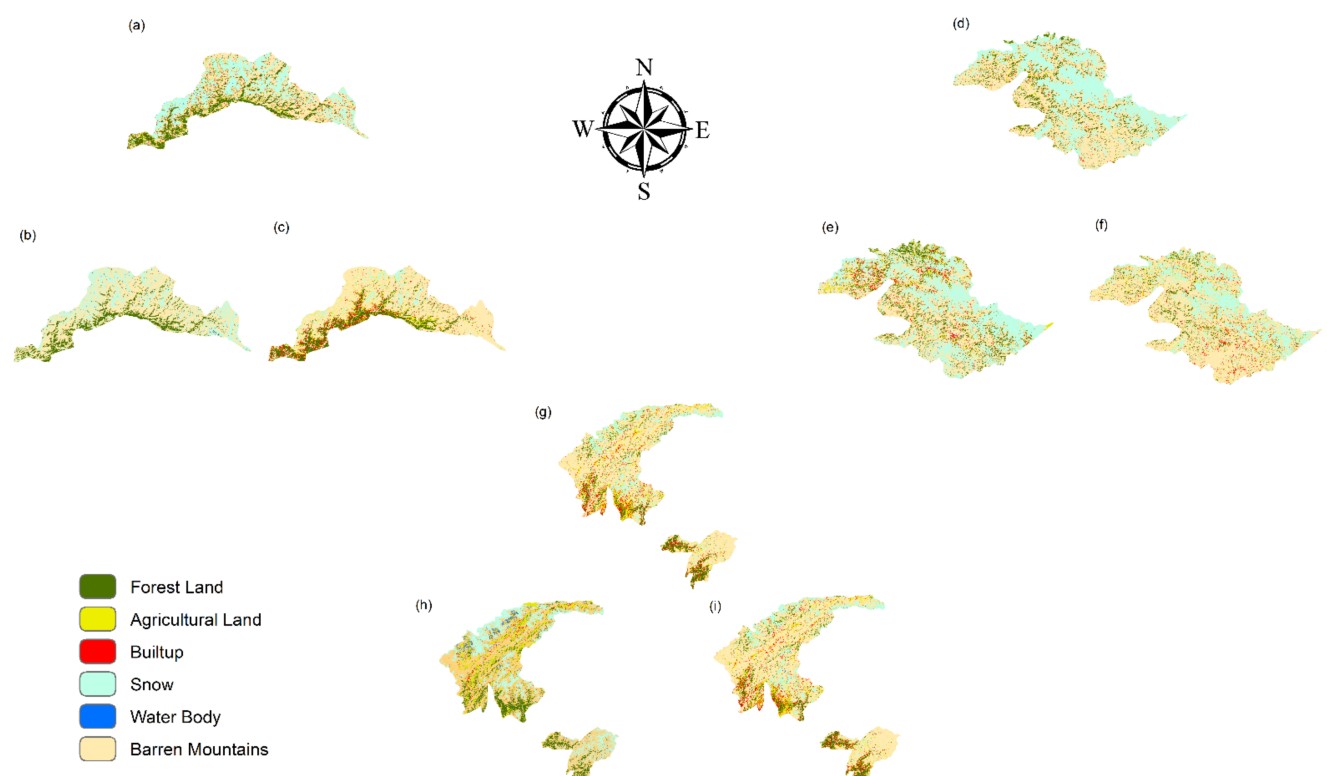

**Figure 4.** LULCC in Azad Jammu Kashmir (AJK) ((**a**) 2000, (**b**) 2010, (**c**) 2020); GB ((**d**) 2000, (**e**) 2010, (**f**) 2020); Khyber Pakhtunkhwa (KP) ((**g**) 2000, (**h**) 2010, (**i**) 2020).

*3.3. LULCC Dynamics in Snow Leopard Range in KP*

The overall dynamics of LULCC in the KP region of Pakistan's snow leopard range is given in Table 5 and Figure 4. A large expansion of 150% in built-up area was observed. Agricultural land and barren mountain area increased by 66% and 32%, respectively. Similarly, the barren mountain area increased by 32%. A loss 69% in snow cover area was observed. Water bodies increased by 29% from 2000 to 2020.

**Table 5.** LULCC Dynamics in Snow Leopard Range in KP.

| Years | Forest Land (km²) | Built up (km²) | Agricultural Land (km²) | Barren Mountains (km²) | Water Bodies (km²) | Snow Cover (km²) | Total (km²) |
|---|---|---|---|---|---|---|---|
| 2000 | 2684.15 | 461.845 | 1224.31 | 10,293.11 | 8.6517 | 6194.15 | 20,866.22 |
| 2010 | 2661.06 | 197.45 | 2182.58 | 11,303.44 | 354.69 | 4167.00 | 20,866.22 |
| 2020 | 2172.42 | 1152.55 | 2033.59 | 13,567.30 | 11.15 | 1929.21 | 20,866.22 |
| Change (km²) | −511.73 | 690.71 | 809.27 | 3274.19 | 2.50 | −4264.94 | |
| % Change | −19 | 150 | 66 | 32 | 29 | −69 | |

### 3.4. LULCC Dynamics in Snow Leopard Range in AJK

The overall dynamics of LULCC in the AJK region of the snow leopard range in Pakistan is given in Table 6 and Figure 4. A 429% increase was observed in the built-up area. An increase of 132% in the agricultural land from 2000 to 2020 was found. An increase of 46% of barren mountain was observed. Forest land was reduced by 19%. Snow cover area was reduced by 85% from 2000 to 2020.

**Table 6.** LULCC Dynamics in Snow Leopard Range in AJK.

| Years | Forest Land (km$^2$) | Built up (km$^2$) | Agricultural Land (km$^2$) | Barren Mountains (km$^2$) | Water Bodies (km$^2$) | Snow Cover (km$^2$) | Total (km$^2$) |
|---|---|---|---|---|---|---|---|
| 2000 | 973.8 | 51.25 | 168.47 | 2405.54 | 104.71 | 1491.95 | 5195.73 |
| 2010 | 786.32 | 112.90 | 187.32 | 2966.99 | 31.63 | 1110.57 | 5195.72 |
| 2020 | 784.30 | 271.02 | 390.35 | 3520.92 | 0.42 | 229.27 | 5196.28 |
| Change (km$^2$) | −189.50 | 219.77 | 221.88 | 1115.38 | −104.29 | −1262.68 | |
| % Change | −19 | 429 | 132 | 46 | −100 | −85 | |

### 3.5. Land Cover Categories Interconversion Dynamics

Differences in land use categories were observed during the study period over time (Figure 5). A total of 15,175.64 km$^2$ snow covered area was converted into built up area, agriculture land, water bodies and barren mountains. An accumulative conversion of 4961.54 km$^2$ barren mountain area converted into forest land, built up area, agriculture land water bodies and snow cover. Only 31 km$^2$ of built up area was converted into agriculture land, water bodies, snow cover and barren mountains. A total of 1428.78 km$^2$ area of agriculture land was converted into forest land, built up area, snow cover water bodies and barren mountains. Only 12.07 km$^2$ of water bodies area was converted into agriculture land, snow cover and barren mountains. Land use categories conversion is presented in Figure 6.

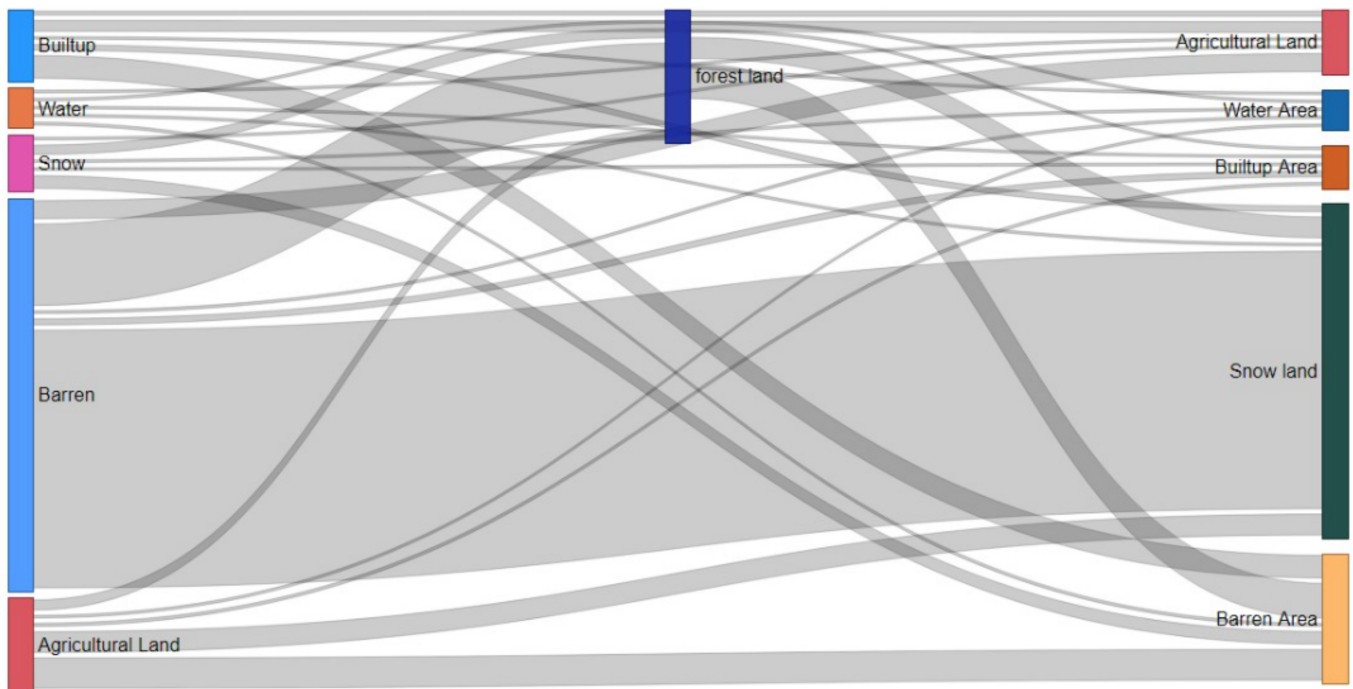

**Figure 5.** The sneaky diagram explicates the proportion of interconversion of LULCC categories.

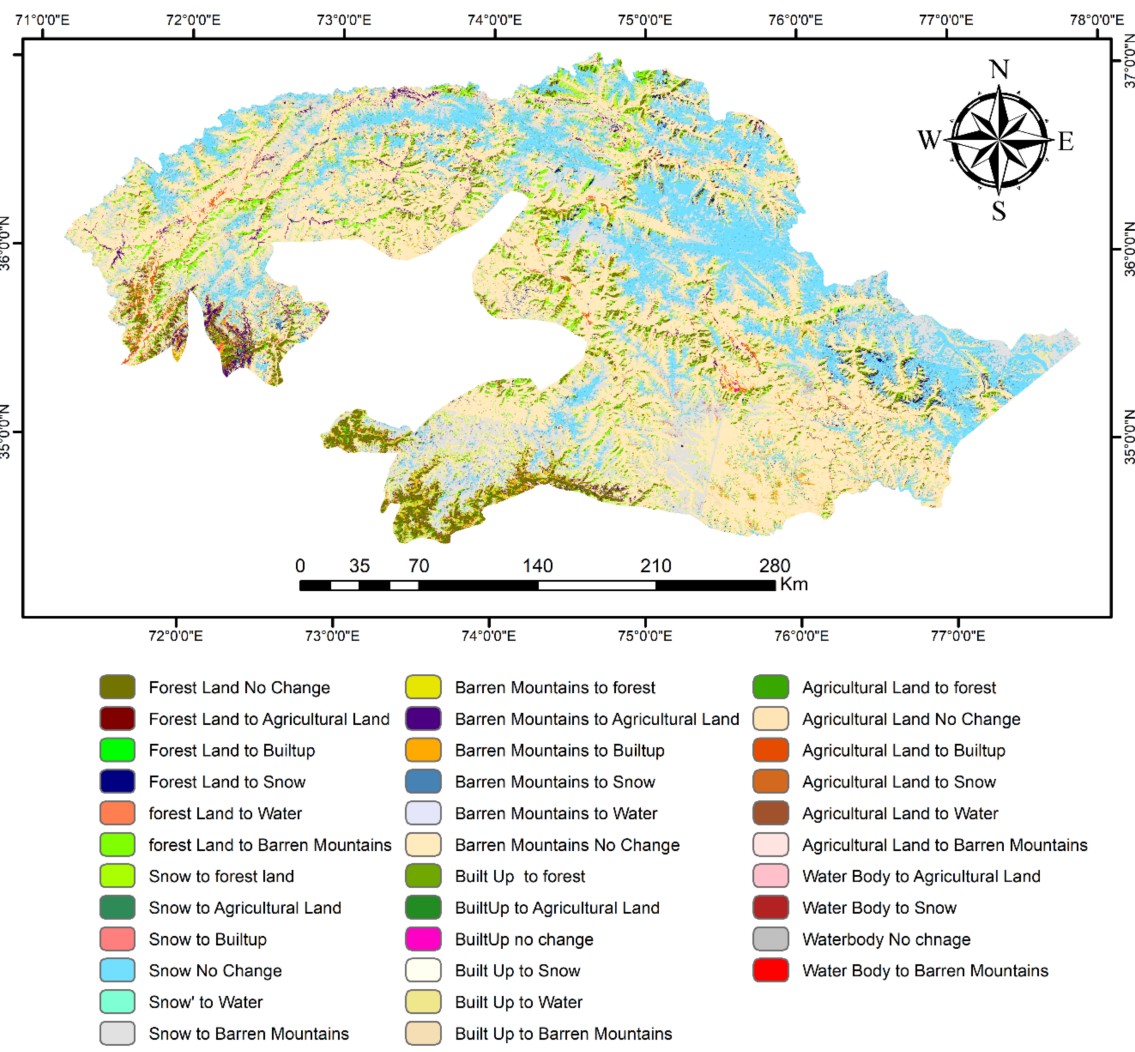

**Figure 6.** Map of LULCC categories interconversion.

## 4. Discussion

In this study, we calculated the LULCC across the entire snow leopard range in Pakistan. We used GIS and remote sensing to quantify LULCC for the years 2000, 2010, and 2020. Remote sensing has been extensively used to measure LULCC to gain useful information and insight into overall ecosystem health [70]. The satellite data used in this study provided adequate spatial variability for LULCC [71]. Our results provide concrete evidence of extensive LULC temporal changes in Pakistan's snow leopard habitat. Assessing the magnitude and rate of these changes also helps to understand the driving factors of LULCC in the study area.

Built-up area and agricultural land expanded by 163% and 153%, from 2000 to 2020, respectively. Snow leopard range in Pakistan spans across GB, KP and AJK, with some areas being densely populated with humans. The observed increase in built-up area and agricultural land could be attributed to a growing human population. Pakistan is the sixth most populous country in the world with a population size of 207.8 million people [50] with a current average annual population growth rate of 2%, it is projected to be the fifth most populous country in the world by 2050 [51]. Human settlements are expanding, and new infrastructures are built deeper into species habitat as land procurement becomes more and more difficult. According to census data from the government of Pakistan, an increase of 36% was reported in the population of AJK (density: 302 people/km$^2$) from 1998 (2,972,523) to 2017 (4,045,366). Similarly, the human population in KP ((density: 300 people/km$^2$)) increased by 72% from 17,743,645 in 1998 to 30,523,371 in 2017 (Figure 7). In GB, the

human population (density: 17 people/km$^2$) increase by 43% from 1998 (870,347) to 2013 (1,249,000) (Figure 8).

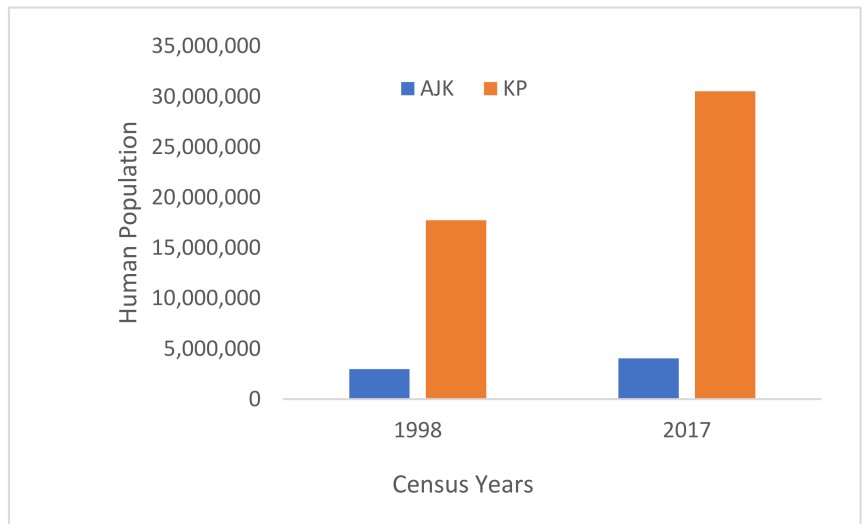

**Figure 7.** Human Population growth in AJK and KP from 1998 to 2017.

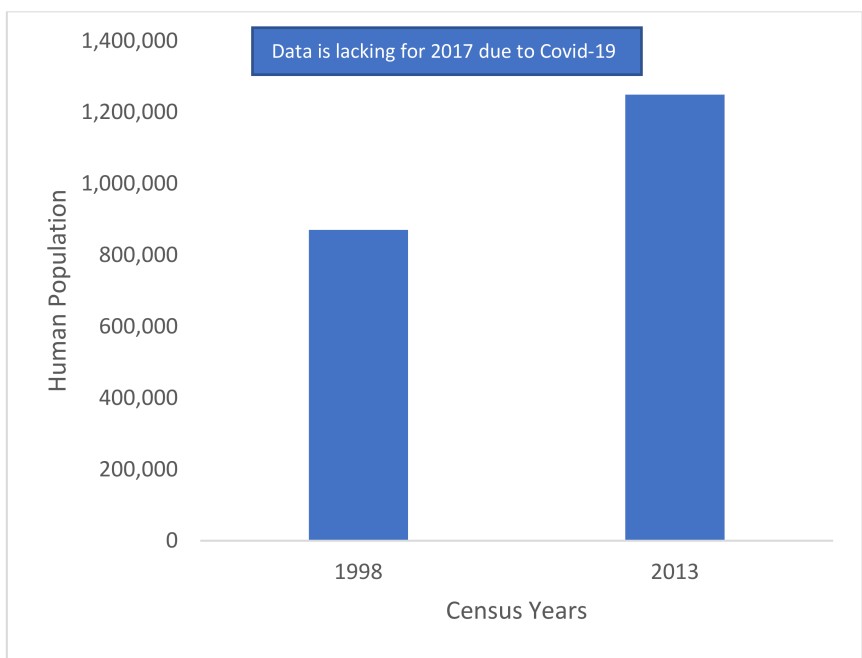

**Figure 8.** Human Population growth in GB from 1998 to 2013.

The growing demand of agricultural products for this increasing human population has intensified the rate of natural terrestrial land being shifted to agricultural land. The rise in this conversion has been exponential in recent decades [72,73]. The increase in agricultural land is typically coupled with an increase in livestock and expansion of grazing grounds deep into species habitat. A greater number of livestock may cause competition between wild and domestic ungulates over high-altitude grazing grounds. Moreover, the situation may make the livestock more vulnerable to snow leopard attacks, which could intensify conflicts between snow leopards and humans [73,74].

We also found that the rate of deforestation and forest loss was exponential [25,72]. Previous studies have concluded similar findings in that rapid urbanization and agricultural expansion are the major drivers of deforestation [74–76]. Forests act as a carbon sink, and

loss of forest area causes the release of more carbon into the environment [77,78]. The rise in carbon emissions due to deforestation and rapid urbanization can also cause subsequent rises in temperature, and a reduction of snow cover as seen in this study [79–82].

In this study a massive decrease in snow cover was observed from 2000 to 2020. This decrease could be attributed to climate change [83–87], and LULCC [88,89]. Snow cover plays an important role in the overall health of an ecosystem [90,91]. It can affect glaciers and the overall hydrology of an area [92]. Northern Pakistan currently has 5218 glaciers covering an approximate area of 15,040 km$^2$ [93]. These glaciers not only feed into rivers but also play a vital role to stabilizing regional and global climates [94]. Unfortunately, these glaciers are already melting at an alarming rate of 0.66 m/year [94] due to human activities and global warming [93]. The decrease observed in water bodies area may be attributed to fewer glaciers feeding into water bodies during the summer months coupled with reduced precipitation in winter months.

Land surface temperature (LST) in the Himalayas, Tibetan Plateau, and Central Asia at large are currently rising at a rate far faster than the global average [95]. Subsequent habitat loss leading to fragmentation of the snow leopard's range will present numerous conservation challenges [96]. A recent study confirmed that parts of snow leopard range in Pakistan have very low habitat suitability for the species [49]. However, this may increase as climate change threatens the mountain landscapes that snow leopards are found in. The Global Climate Risk Index has placed Pakistan fifth on the list of countries most vulnerable to climate change in its annual report for 2020 [52]. Very little work on how climate change may impact ecosystems in Pakistan has been done [97]. One study forecasted a mean temperature increase of 3.8 °C by 2100 [98]. Another study claimed a potential increase of 1.4 °C to 3.7 °C by the 2060s, and a potential increase to 6.0 °C by the 2090s [99]. In the last 50 years, the annual mean temperature in Pakistan has increased by roughly 0.5 °C [100]. Species such as the snow leopard will likely face more dire consequences, as the northern high-altitude regions of the country are expected to warm faster and at a higher rate [99].

Our finding of massive LULCC across snow leopard range in Pakistan is not an isolated occurrence. Snow leopard distribution has contracted across its entire range [101]. Inside and near protected areas, the threat to snow leopards from habitat degradation and fragmentation is increasing due to livestock grazing, forest clearing for agriculture and pasture, and the collection of medicinal and aromatic plants [102]. The observed expansion in built-up area and agricultural land could cause further habitat reduction and fragmentation. An increase in built-up area is usually associated with an increase in overall infrastructure, including new houses, settlements, residential and commercial buildings, pipelines, and roads. This causes a direct loss of species habitat. Roads can bisect populations and reduce gene flow and thus genetic diversity. In addition, roads provide easier access for humans to reach snow leopard habitat, which may escalate the rate of wildlife trafficking, illegal hunting, poaching, and pollution. Collisions of snow leopards with vehicles may also be fatal. While there is currently little research to suggest that roads at present are having negative impacts on snow leopards [103] the construction of major throughways, such as the China–Pakistan Economic Corridor (CPEC) [104,105] will likely increase impact of roadways on wildlife. Projects like the CPEC will directly affect snow leopards by fragmenting and degrading their habitat. It will also contribute to localized warming, as it is estimated that approximately 7000 trucks will pass through this area daily during the operational phase, leading to the emission of up to 36.5 million tons of $CO_2$. These emissions could drastically reduce snow-covered area and will negatively impact glaciers [106].

Given the potentially dire outcomes associated with LULCC observed in this study for snow leopards, we suggest several conservation actions. First, continuous and consistent long term monitoring of LULCC, their causes, and direct and indirect outcomes for wildlife like the snow leopard should be established. Consistent monitoring may help to better predict outcomes associated with LULCC and will aid in quickly addressing newly emerging conservation challenges. Monitoring should consist of a series of extensive surveys and

open communication between administrative districts in Pakistan to share information and work collaboratively.

Second, creating protected areas is the best way to ensure the conservation of a species or ecosystem. Approximately, 24% of the snow leopard habitat is laying inside protected areas of different sizes. According to a recent study [49] most of the suitable habitat of snow leopard in Pakistan has already been protected, however there are some areas presenting suitable habitat are outside of any declared protected area. The same study concluded that most of the national parks had weak links with regards to movement of snow leopard across different habitats. To mitigate or minimize LULCC, protected area networks should be strengthened throughout snow leopard range in Pakistan. In addition to managing pre-existing protected areas, new protected areas should be developed to protect currently suitable habitat from quickly expanding built-up area and agricultural land. The government should declare buffer zone areas for existing national parks and should strictly follow wildlife protection laws to ban developmental and agricultural practices in and outside of these zones. A model for this practice is the GB wildlife department, which is effectively protecting habitat in Khunjerab National Park (KNP). These same management strategies should be implemented in other protected areas. To ensure the implementation of wildlife laws, capacity building of forest and wildlife department staff is needed. In addition, staff should be logistically supported to protect species in this rough and rugged terrain.

Third, the government and NGOs should initiate projects on sustainable community based natural resource management, sustainable livelihood practices, and solutions to reduce human-wildlife conflicts. Literature suggests that formal and informal education helps to increase public understanding and acceptability of wildlife and is an effective solution to dilute people's hatred for predators [107,108]. It increases public understanding and plays a key role in equipping people with pro-conservation attitudes and practices. Initiating community learning sessions, engaging youth in conservation, and organizing other awareness raising events will help to change the perception of locals towards snow leopard. Community-based surveillance that monitors snow leopard habitat and prevents wildlife crime should be implemented to protect the snow leopard from illegal hunting that may results as humans gain easier access to snow leopards.

Fourth, the government should implement forest protection laws to counteract illegal forest harvesting. Most of the communities living inside snow leopard range in Pakistan rely on the forest for their domestic and commercial needs. The government should provide alternative resources to locals that reduce their dependency on forest harvesting. The use of renewable and energy efficiency systems for cooking and heating should be introduced and encouraged to reduce pressure on forest and range land. Community based forest and land protection efforts could be very helpful to conserve and sustainably use natural resources within snow leopard range. Planting micro-forests at community levels should be practiced to combat rising $CO_2$ levels that lead to an increase in land surface temperature and consequently loss of snow-covered area and snow leopard habitat.

Fifth, countries bordering the study area that are making large scale trans-boundary infrastructural changes should work with Pakistan in following national and international levels of environmental protection protocols. A comprehensive Environmental Impact Assessment study should be carried out before starting any developmental project in the area. The government should also monitor overall developmental projects inside the study area, so that adverse effects on snow leopard habitat can be minimized.

## 5. Conclusions

Snow leopard range in Pakistan supports the world's third largest snow leopard population (250–400 individuals, tied with India) throughout its 12-country range [109]. It is an important wildlife corridor for genetic flow and dispersal, as it is shares borders with snow leopard habitat in China, India, and Afghanistan. Therefore, any damage to snow leopard habitat in Pakistan would be consequential for the species across its entire. In this study, LULCC were identified and measured across snow leopard range in Pakistan

using GIS and remote sensing data. Our results showed a massive expansion in built-up area and agricultural land from 2000 to 2020. An increase in the barren mountain area and decrease in forest cover and snow-covered area was also recorded during the study period. The expansion of agricultural land and built-up area could be attributed to a massive increase in the human population across snow leopard range in Pakistan, while the loss of snow-covered and water bodies area could be due to increasing land surface temperature because of climate change. Based on findings of this study we recommend conducting a series of extensive surveys to better understand how the negative impacts of LULCC can be mitigated, an increase in protected areas, support of sustainable living practices for residents, tighter legislation surrounding wildlife and forest protection, and increased cooperation among countries sharing borders with Pakistan within snow leopard range. These actions would help to minimize the harmful impact of LULCC on the quantity and quality of snow leopard range in Pakistan.

**Author Contributions:** Conceptualization, T.U.K. and X.L.; Data curation, T.U.K., M.A.S., M.C., C.Z. and E.U.D.; Formal analysis, T.U.K. and A.M.; Funding acquisition, X.L.; Investigation, T.U.K., A.M., E.U.D. and B.U.K.; Methodology, T.U.K., A.M. and C.E.H.; Project administration, T.U.K., X.L. and S.A.; Resources, X.L., A.M. and M.N.; Software, T.U.K., A.M. and M.A.S.; Supervision, X.L.; Validation, A.M., B.U.K., E.U.D., M.C. and C.Z.; Visualization, A.M., T.U.K. and M.A.S.; Writing—original draft, T.U.K.; Writing—review and editing, X.L., C.E.H. and M.N. All authors have read and agreed to the published version of the manuscript.

**Funding:** This study was jointly supported by grants from The Ministry of Science and Technology of the People's Republic of China (research and application of key techniques on endangered species conservation and prediction of forest fire and pests in response to climate change, 2013BAC09B00) and National Forestry and Grassland Administration (management and improvement of monitoring in national parks, 2018HWFWBHQLXF-01).

**Institutional Review Board Statement:** Ethical review and approval were "Not applicable" to this study, because it was not involving any humans or animals handling.

**Informed Consent Statement:** Not applicable to this study.

**Data Availability Statement:** There is no supplementary data.

**Conflicts of Interest:** The authors declare no conflict of interest. The funders had no role in the design of the study; in the collection, analyses, or interpretation of data; in the writing of the manuscript, or in the decision to publish the results.

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
