# Peer review of "Use of GIS and Remote Sensing Data to Understand the Impacts of Land Use/Land Cover Changes (LULCC) on Snow Leopard (Panthera uncia) Habitat in Pakistan"

_sustainability, doi:10.3390/su13073590_

Round 1

Reviewer 1 Report

The paper deals with permanently current topic, which is not strictly concerned only the observed area (region of Pakistan) and only the specific animal species (snow leopard), but this issue is problematic in several regions among the world and is related with many animal species. Although the topic is very interesting and important, several parts of the paper should be improved and completed.

The aim of the research should be defined strictly and clearly in the Introduction.

Authors should answered these main questions:

Why did they choose years 2000,2010,2020? Are these years specifical for some reason or this selection was made only for the accessibility of the satelite images?

Snow leopard Global District Range was delimited according to IUCN’s Red List of Threatened Species website. What is the year of the delimitation? Are the borders still actual? Maybe, authors should disscuss the delimitation of this area in Disscussion.

Authors use the ENVI software – it would be appropriate to explain or define, what the software is usefull for, why they chosen it for they research. In the line 179 – authors are writting about ENVI 5.1 and in the line 207 about ENVI 5.3. Is it just writting error or is there any difference between these two versions?

The paper deals with snow leopard habitat, but this habitat is not described in detail. What does the snow leopard really need? Forest or mainly the area with snowed cover? It is important for the study.

Authors should also engaged in some research of increase or decrease of snow leopard in last decades. What is the actual population of the snow leopard in the observed area? In the line 405, the attacks of the snow leopards are mentioned, but citation is missing. Are there any evidence of the conflicts between people and leopards?

The protection of he observed area is mentioned firstly in the line 368 or 449. This should be also mentioned in the description of the area (2.1 Study area), if there is some degree of landscape protection (e.g. national parks). What area is protected and how?

Line 249 – What is the reason of the decrease of the water bodies in the observed area?

Lines 240-250 – it is not neccessary to write about all area changes with km2 – text is disorganized – all needed numbers are clearly stated in the table 3. Authors should focused on reasons of changes.

Line 239 – 1,103.58 km2 – subscript should be changed on superscript

Figure 2 – why are missing charts about water bodies and Barren Mountains?

Figure 3, Figure 4 – small illegible maps

This type of research should also contain an analysis of change type (character) – what land cover category changed to another land cover category (for example, it is usefull to know if the forests were changed to built-up area or to the agricultural area) – then, we are able to identify the driving forces of changes. This analysis should be based on the map of stable or unstable areas. When you know, where the anstable areas are, you can focus the level of landscape protection to this area.

Chapter 3.2, 3.3., 3.4 – what is the reason of detailed land cover analysis for the three administrative units? I think, that it would be better to describe and analyse land cover changes in entire area. I dont know, if there is any close relation between snow leopard distrubution range and administrative borders. 

Reviewer 2 Report

The current study is an important one in terms of land management. Management of natural habitat for biodiversity is a major issue in Pakistan. Furthermore, changing climatic conditions and anthropogenic activities are also deteriorating the conditions for snow leopard. Although the study is well conducted. However, some major deficiencies must be overcome before its processing. I suggest a major revision for this article (sustainability-1147608-peer-review-v1).

  • Systematic abstract is missing. An abstract must contain an introduction, identified problem with aim of the study, quantitative results, and conclusion with future recommendations. Please give a conclusive conclusion with a future perspective at the end of the abstract.
  • The lack of a clear-cut hypothesis is a major drawback of this study. Without any hypothesis, the research article always remains incomplete. Please provide the hypothesis at the end of the introduction.
  • No statistical analysis is provided on survey data. Authors must have to apply some statistical model or analysis to improve the quality of survey data. Also give some details about statistical analysis done in the study by providing a sub-heading in the material and methodology section.
  • Authors must have to use SI units’ symbols. In SI capital K is used for kelvin. For kilo, they have to use small k throughout the manuscript.
  • Discussion part has many unnecessary information. Authors must focus on those attributes which they have study and the mechanism behind them.

Round 2

Reviewer 1 Report

I think, that all comments and suggestions were incorporated very well and current version of manuscript is clearly arranged and well prepared for publishing. Pictures or maps which were added to the manuscript improved its quality.

Reviewer 2 Report

The manuscript has been substantially modified, According to me, it is now ready for publication with no further revision. I recommend accepting the manuscript in the present form.